# Assessment of Meat-Type Sheep Welfare Using Animal-Based Measures

**DOI:** 10.3390/ani11072120

**Published:** 2021-07-16

**Authors:** Naceur M’Hamdi, Cyrine Darej, Khaoula Attia, Hajer Guesmi, Ibrahim El Akram Znaïdi, Rachid Bouraoui, Hajer M’Hamdi, Lamjed Marzouki, Moez Ayadi

**Affiliations:** 1Research Laboratory of Ecosystems & Aquatic Resources, National Agronomic Institute of Tunisia, Carthage University, 43 Avenue Charles Nicolle, Tunis 1082, Tunisia; cyrine.darej@gmail.com (C.D.); attiakhaoula@ymail.com (K.A.); guessmihajer55@gmail.com (H.G.); 2Department of Animal Sciences, High Agronomic Institute of Chott Mariem, University of Sousse, Sousse 4000, Tunisia; akram_znaidi@hotmail.com; 3Laboratory ADIPARA, Higher School of Agriculture of Mateur, Road Tabarka-7030, Mateur, Bizerte 7030, Tunisia; bouraoui.rachid@yahoo.fr; 4Ministry of Agriculture, CRDA Ben Arous, New Medina, Ben Arous 2063, Tunisia; hajervet@gmail.com; 5Unit of Functional Physiology and Bio-Resources Valorization (BF-VBR), Higher Institute of Biotechnology of Beja, University of Jendouba, Beja 9000, Tunisia; Lamjed.marzouki@ipeis.rnu.tn (L.M.); moez_ayadi2@yahoo.fr (M.A.)

**Keywords:** animal-based measures, indicators, sheep welfare, stress

## Abstract

**Simple Summary:**

There was every indication that animal welfare will continue to be a major issue affecting livestock farming in the future. The main welfare issues affecting sheep were feeding strategies, health, and diseases. The health problems of sheep are avoidable with good grazing, breeding, and stockmanship. However, sheep must be given adequate supervision to ensure that any welfare issues are quickly noticed and addressed. Assessing animal welfare can be used as management tools by farmers to identify welfare issues and recognize poor welfare.

**Abstract:**

This study aimed to assess the welfare of Tunisian sheep in extensive sheep production systems using animal-based measures of ewe welfare. This study encompasses the first national survey of sheep welfare in which animal-based outcomes were tested. Animal-based welfare measures were derived from previous welfare protocols. Fifty-two Tunisian farms were studied and a number from 20 to 100 animals by flock were examinated. The whole flock was also observed to detect clinical diseases, lameness, and coughing. The human-animal relationship was selected as welfare indicators. It was evaluated through the avoidance distance test. The average avoidance distance was 10.47 ± 1.23 and 8.12 ± 0.97 m for a novel person and farmer, respectively. The global mean of body condition score (BCS) was 2.4 with 47% of ewes having a BCS of two, which may be associated with an increased risk of nutritional stress, disease, and low productivity. Ten farms had more than 7% of lambs with a low body condition score, which may be an indication of a welfare problem. The results obtained in the present study suggest that the used animal-based measures were the most reliable indicators that can be included in welfare protocols for extensive sheep production systems.

## 1. Introduction

Sheep farming in Tunisia occupies an important place in the economic and social levels. It is the main source of income for most of the rural population [1]. The national sheep flock size accounts for 3.7 million heads of sheep and contributes to around 42.5% of the red meat and 5% of milk production [2]. However, market demand from consumers for assurance schemes for high-quality and safe animal products is increasing [3,4]. The concept of welfare in animals has gained importance in recent years. That is due to the fact that ensuring animal welfare is not only a duty that has to be performed legally and ethically but it should also be considered as a way of direct economic contribution to the enterprise. The concept of welfare comprises physical and mental health [5]. The welfare of animals means a life away from any undesired emotions (pain, suffering, and distress). On-farm welfare assessments can be used for immediate or ongoing on-farm monitoring and to demonstrate compliance with national and international legal welfare standards and farm assurance schemes [6,7]. Parameters used for the assessment of animal welfare can refer to either the physiology, behavior, production or health of an animal [8]. Therefore, the objective of this study is to evaluate the welfare state of Tunisian extensive sheep using selected animal-based welfare measures.

## 2. Materials and Methods

### 2.1. Study Sites and Animals

This study was carried out in the North of Tunisia in the sub-humid bioclimatic area (rainfall > 550 mm). Fifty-two farms were visited during the lambing season (September–December) in 2017. They were selected through random sampling from lists of farmers data obtained from the Northwest Forestry and Pastoral Development Office of Beja, Tunisia. The farmers were contacted and asked whether they wanted to participate in the study. A total of 1040 Noire de Thibar ewes (840) and lambs (200), aged 2–6 years, from small and medium flocks of approximately 350 breeding ewes were randomly selected using systematic random sampling and examined by a one trained person at lambing, mid lactation, and weaning. Each farm was visited three times during the period of the study. The ewes were managed under extensive conditions, in a year-round outdoor system, grazing pastures and managed under commercial conditions. Natural rangelands forage (hay, barley in green, etc.) and concentrated feed were the main food resources for sheep. The ewe sample size was selected based on the following equation reported by Cochran [9] under a 95% confidence interval and precision of 10%. This number was supported by the Animal Welfare Indicators Project (AWIN) sheep protocol, which recommends a sample of 92 animals when the farm size is ≥2000 breeding ewes [10].
$$n=\frac{N}{1+N{\left(e\right)}^{2}}$$
where *n* is the sample size, *N* is the population size, and e is the level of precision.

The flocks were observed to detect signs of clinical disease, lameness, and coughing. The human-animal relationship and fear testing were selected as welfare indicators based on behavior at pasture. 

### 2.2. Welfare Indicator Assessments

Indicators used in this study, were inspired by AWIN [10] (Table 1). Groups of sheep, ranging in number from 20 to 100, from each selected study farm were presented by the farmer and assessed using eight indicators of welfare [11,12,13]. These indicators were considered to be key animal-based outcome measures to be included in the on-farm protocols [11]. The validity and feasibility of the indicators selected in this study have been previously justified and reported in other studies [13,14,15]. The body condition score was used as an indirect measure of good feeding. The body condition was scored on a scale from 1 to 5 [16] during lambing and at mid-lactation for ewes and weaning for lambs. A score of 1 is very thin and 5 is very fat. Good housing was evaluated through fleece cleanliness [10]. All the animals were assessed and scored on visual cleanliness of the fleece (0–3 scale). The score 0 represented a visually clean fleece, with minor fecal material or mud in the fleece. A score of 1 represented small spots of dirt under the belly, legs, and tail; a score of 2 represented a generally dirty fleece; and a score of 3 represented a very dirty fleece, stained with fecal material or mud under the belly, legs, and tail. For good health, three indicators were evaluated; lameness, respiratory disorders, and dirtiness [10]. Lameness was scored following a 3-point scale (0–2) that takes into account the smoothness of movement (score 0 for imperfect mobility, score 1 for lame, and score 2 for severely lame. The respiratory rate was determined by measuring the time (seconds) required to take ten breaths; these data were then converted to breaths per minute (bpm).The respiratory rate was scored on a scale from 1 (acute) to 3 (progressive). Dirtiness was evaluated using a 3-point scale, through a visual assessment of one side and behind the hindquarters and belly. The human-animal relationship (HAR) was defined as the degree of relatedness or distance between the animal and the human. The test was conducted by the trained person and the farmer according to the procedure reported by Waiblinger et al. [17]. The heart rate was measured electrocardiographically. In short-term measurements of 1 h on restrained standing animals two skin electrodes were adequate, placed on each side of the thorax in the plane of the heart [18]. Each indicator was assessed by observing the behavior and physical appearance of the individual sheep within the group and scored. Following each assessment, the observer recorded the number of sheep observed with each welfare indicator (count data). The total number of sheep in each assessment group was counted to determine the number of animals not affected by each welfare condition.

### 2.3. Statistical Analysis

Data analysis was performed using the SAS statistical package [19]. A descriptive exploratory analysis was carried out to summarize the main characteristics of the assessments performed. The overall proportion (%) of affected sheep recorded was calculated by dividing the total number of affected animals by the total number of sheep in the sample group or the flock. The significant difference between proportions was calculated by chi-square. Results were expressed as means ± SD. Results with an associated probability less than or equal to 0.05 were considered significant.

## 3. Results 

### 3.1. Body Condition Score

Assessing body condition was an important animal-based measurement. Descriptive statistics for BCS were presented in Figure 1. More than 75% of the BCS of ewes recorded in the study ranged from 2.5 to 3.5 at lambing. Ewes were considered lean if the score was less than 2 and fat if the score was equal to 5. In our study, we noticed that 17% of lambs and 24% of ewes had a body condition score of less than 2. Therefore, the highest number of animals (57 and 64%) had a BCS of 3. However, only 2.5% of ewes were considered fat.

### 3.2. Cleanliness

For good housing, a flock can be considered clean, since 60% of ewes had a cleanliness score of 1 and only 14% were considered dirty (Figure 2). There was no significant difference between lambs and ewes (*p* > 0.05), this can be explained by the fact that during our visits to farms, lambs were not separated from their mothers.

### 3.3. Health

Good health was an important component of animal welfare and it can be defined as the absence of injuries, disease, and pain [20]. For health, four indicators were assessed: Respiratory rate, lesions and dirtiness, and finally lameness (Table 2). We reported in our study an average percentage of moderate lameness ewes with values of 10.5% and 4.65% with severe lameness. Lameness was considered the most common sign of limb injury, which compromises the animals’ welfare. In our study, the significant percentage of sheep with no lameness (84.87%) may be indicative of good overall welfare within the flock. For the respiratory rate, 89.75% of animals had no respiratory problem and only 8% had a moderate problem.

### 3.4. Animal-Human Interactions

The quality of the human-animal relationship can be one of the most important factors in determining the welfare of an animal. The nature and frequency of this relationship can vary markedly in different sheep farming systems. The mean withdrawal distance was all greater (*p* < 0.01) for a novel person (10.47 m) than the farmer (8.12 m) (Table 3). Moreover, sheep withdrew from the advancing person at distances exceeding 20 m in the approach. For heart rates, the average value was 128.4 bpm for a novel person for a test during the 10 s. HAR was significantly (*p* < 0.05) higher compared to the HAR reported for the farmer (97.8 bpm).

## 4. Discussion

### 4.1. Body Condition 

This study constitutes the first evaluation of the welfare of sheep conducted in Tunisia. The body condition score assesses the amount of fat and muscle overlying the spine. Overall, the body condition was considered good. Previous studies have also shown similar results [21]. In other studies [22,23], the general average score for the body conditions of Norduz ewes was 2.9. The descriptive analysis of BCS reveals that the median of BCS was 3.0 and only 5.5% of animals were scored with a BCS lower than 2 and 15% greater than 4. These results agreed with those of Keinprecht et al. [24]. Most of the ewes in this study were within the recommended body condition. However, thin ewes were observed within flocks, suggesting that some farmer’s nutritional management was not and or were not identifying/treating individual thin ewes. A low BCS, at mid-lactation indicated prior long term poor welfare. In addition, low values of BCS occured when energy expenditure surpasses the intake and body fat was mobilized to meet the animal’s needs, whereas high values indicated over-feeding. Indeed, ensuring that all sheep in a flock meet their nutritional requirements was not easily achievable in extensive systems.

### 4.2. Cleanliness

Fleece cleanliness had previously been proposed as an important welfare measure for sheep, as it can provide information about the quality of the environment. In our study, animals were reared in simple shelters and a large backyard. A higher score for fleece cleanliness was reported in this study (Figure 2). Previous studies [15,25] reported similar results and explained the higher score to the good housing conditions. Several authors [8,15,26] suggested that looking at the degree of dirtiness/cleanliness of a sheep in a flock can give a good insight into the housing conditions. However, in an Italian study [14], authors assessed animals and found significant differences between lambs and ewes.

### 4.3. Health

It was well acknowledged that health and disease were important aspects of welfare. The indicators selected in this study were respiratory disorder, lameness, and lesions. Lameness is one of the major welfare concerns of sheep. Lame ewes were found across all farms, while moderate and severe lameness were notified to farmers. In our study, the significant percentage of sheep with no lameness (84.87%) may be indicative of good overall welfare within the flock. However, Winter and Arsenos [27] reported a high prevalence (up to 75% of sheep). For respiratory disorder, we noticed that 8% of ewes have moderate disorders and 2.5% with a severe disorder which is considered higher than other findings [12]. For the respiratory rate, our results agreed with those of Lees et al. [28]. For lesions, we reported 86.95% with no problem and 57% with moderate lesions. We can conclude that the flocks required more care and attention to avoid health problems, lesions, and injuries [12,25,29].

### 4.4. Animal-Human Interactions

The assessment of an animal’s reaction to humans was a good indicator of the human-animal relationship. In this study, the presence of the farmer had a strong calming effect (Table 3). Our results were highly consistent with previous results that affirmed that the presence of a familiar person can calm the animal [30]. Furthermore, the presence of a familiar person reduced stress and fear of humans in sheep [31,32]. However, for a novel person, sheep showed a higher avoidance distance [26]. HAR was a good indocator of the stockpersons’ attitudes towards farm animals. Moreover, the low average approach distance of our study was explained by the fact that sheeps recognize individual humans and are more likely to approach those who treat them well than those who act in an aggressive way. Then, the presence of a familiar human may calm the animals in potentially aversive situations. HAR was a major determinant of sheep welfare and particularly pertinent to extensive systems with limited interactions with people. The heart rates of the sheep in this study were similar to the previous values [5,33]. In other studies, higher heart rates in Scottish Blackface lambs were found [8]. The attitude a stockperson holds about animals will strongly influence their behavior towards animals [34]. Conversely, the regular experience of positive human-animal interactions can decrease the animals’ general level of stress [35] and enhance the reproductive performance and the presence of a familiar person can calm the animal in potentially aversive situations [30]. The reaction to an approaching human may be best suited for use when assessing extensively managed animals as it most closely resembles the situations the animals experience regularly.

## 5. Conclusions

This was a first and preliminary study in Tunisia. We can conclude that on-farm welfare assessments can be used for immediate or ongoing on-farm monitoring by farmers. Animal-based measures often reflect the outcome of resource inputs and management practice. The use of behavioral principles should improve the efficiency of livestock handling and reduce stress on animals. The present study reveals the extent to which this species is capable of habituating to common human-related stimuli. The respiratory rate was considered a serious welfare consequence for lambs. The human-animal relationship (HAR) is a major determinant of sheep welfare since it is an important source of fear in farmed sheep.

## Figures and Tables

**Figure 1 animals-11-02120-f001:**
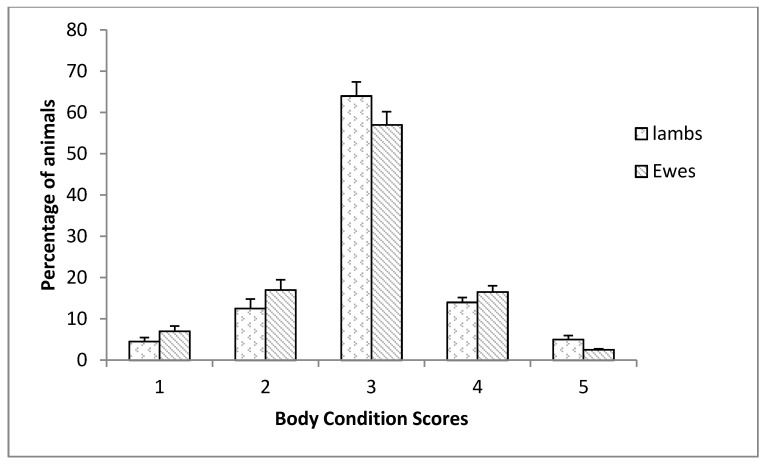
Distribution of the average scores of body condition for the studied sheeps at lambing, mid-lactation, and weaning.

**Figure 2 animals-11-02120-f002:**
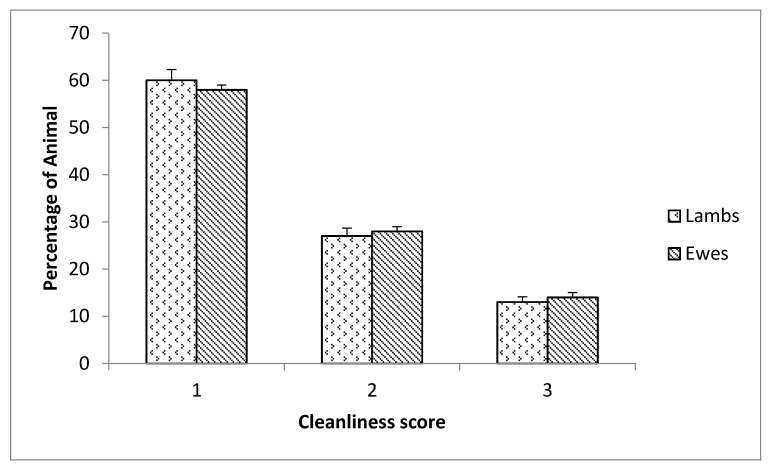
Distrubition of the average of scores of cleanliness for the studied sheeps at lambing, mid-lactation, and weaning.

**Table 1 animals-11-02120-t001:** Animal welfare indicators for sheep adopted from AWIN (2015).

Welfare Principles	Welfare Criteria	Welfare Indicators
Good feeding	Appropriate nutrition	Body condition score
Good housing	Comfort around resting	Fleece cleanliness
Good health	Absence of diseases, injuries, and pain	Lameness, respiratory disorders, and rear and dirtiness
Appropriate behavior	Expression of social behavior	Familiar human approach

**Table 2 animals-11-02120-t002:** Mean ± standard deviation (SD) of the selected health indicators measured in 52 farms for ewes at lambing and mid-lactation.

Variables	Percentage of Ewes with		
	No Problem	Moderate	Severe
Respiratory rate	89.75 ± 7.24	8.1 ± 0.21	2.15 ± 0.14
Lesions	86.95 ± 6.13	9.7 ± 0.33	3.35 ± 0.23
Dirtiness	17.2 ± 3.2	57.3 ± 5.17	12.75 ± 1.33
Lameness	84.87 ± 5.27	10.48 ± 1.12	4.65 ± 0.57

**Table 3 animals-11-02120-t003:** Mean ± SD of avoidance distance and heart rate measured in 52 farms for ewes at lambing and mid-lactation.

Approach Test	Withdrawal Distance (m)			Heart Rate (bpm)		
	Min	Mean	Max	Min	Mean	Max
Novel person	7.5	10.47 ± 1.23	13.78	63.5	128.4 ± 1.42	208.7
Farmer	5.24	8.12 ± 0.97	10.17	58.3	97.8 ± 6.45	197.6

## Data Availability

Data will be published under demand.

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
