# Peer review of "Assessment of Meat-Type Sheep Welfare Using Animal-Based Measures"

_animals, 2021, doi:10.3390/ani11072120_

Round 1
Reviewer 1 Report
The presented study seeks to investigate animal-based welfare indicators for extensive sheep production in Tunisia. The presented data show results of a national survey investigating the proposed indicators in 55 herds adding new knowledge to the welfare of sheep kept in extensive production systems.
However, I have some major concerns about the study in its present form. Firstly, the data presented do only partly comply with the stated objective of evaluating/testing welfare indicators. If the objective is to evaluate the chosen indicators, the validity of the proposed protocol should be validated against an acknowledged protocol. Hence, the selection of measures and scores should be justified, tested (IOR) and validated on-farm. If, on the other hand, the objective is to evaluate the welfare state of Tunisian sheep, the objective needs to be rephrased. The results stated point towards the latter objective.
If the AWIN protocol served as inspiration, how did the chosen indicators reflect the four principles of Good Feeding, Good Housing, Good Health and Appropriate Behaviour?
There is no comprehensive description and justification of the choice of indicators used in this study. Why did you choose the eight indicators? Why did you only choose animal-based measures and no additional resource-based measures?
Furthermore, indicator scales and definitions of levels are very sparsely described and should be elaborated more. As indicator levels are presented now, it is hard to regard these as objective, e.g. lines 67-71 describing scores for fleece cleanliness. How large areas should be affected for the given score levels? A term like " generally dirty fleece" does not sound very objective. I do not understand the indicator "rear" (line 72)?
How was the approach test performed? Were lambs and ewes tested both at the same time?
Another important point is the sampling of herds and animals. How did you derive at a sample size of 55 herds? What sample size calculation did you use? Only herds in the northern part of the country were selected, but no information on the national population is given in order to understand this selection. Finally, having farmers sample animals poses a potential selection bias, and questions the representativeness of the present study. Why was the sampling procedure from the AWIN not followed?
The M&M sections mentions statistical analyses, but it is not clear, what hypotheses were included.
The discussion section mentions heart rate (HR) measures (line 187) - was this measured in the present study (and how)?
In the conclusion lines 187-188 you state, that the present study shows how sheep are able to habituate to human-related stimuli. How can you conclude this based on the HAR test? Furthermore, the respiratory disorders in lambs have not been mentioned in the results section where only lameness is highlighted. Thus, the results and conclusions should be harmonized.
The manuscript needs another proof reading for grammatical errors. In general past tense should be used consequently in the M&M, result and conclusion section. A few corrections are mentioned here:
- line 27 missing . and rephrase to "...more than 7 % of lambs..."
- line 49 missing part of sentence
- lines 72-74 use a consistent terminology with either capital letters or quotation marks
- line 96 Figure with capital F
- line 129 missing space
- line 167 "found" not founded
- line 168 missing part - "information"?
- line 199 "rights" not wrights
- line 231 "Keeling" not Keeliong
Author Response
the answer is in a separate paper

Reviewer 2 Report
Review of Dairy sheep welfare using animal-based measures
Manuscript ID: animals-1123116
A reasonable paper that describes welfare assessment in on 50 Tunisian Farms. The paper is easy to follow however there several areas that the authors need to address.
- There needs to more details as to management of the flocks. Are they all dairy ewes and are the same breeds of sheep found on the 50 farms? What is the management of the ewes i.e. it seems they are housed for a period - is this night or winter - none of this is clear in the material and methods?
- In the introduction mention is made of flock contribution to meat production yet title is dairy sheep. What is dairy production from sheep in Tunisia.
- Observations for signs of clinical disease, was this done by a veterinarian ( or trained person) was every sheep observed or just the 20 on one occasion (or multiple occasions) at what stage of the production cycle were sheep at when observed what was the pasture conditions (range conditions considering it was extensive sheep production) at the time (i.e. dry season wet season etc.)
- Welfare indicators – was it the same person condition scoring, and measuring other welfare indicator scores on each farm if not how did you standardise the between operator bias
- There are no standard errors to the means in the paper - this needs to be there to indicate variability
- In discussion and table 3 mention is made of the farmer and a novel person yet there is no mention of that in the material and methods was it the same novel person each time on all farms ? was it done just once or several times and at what time of year in relation to the production cycles were measurements taken
- Under health 4.3 need to be more explicit re respiratory disorders - there are several causes of this - was a diagnosis or sample get taken to diagnosed
- Re heart rate how was it measured and how often and who measured and was it the same person on each farm
- Re tables and figures the captions are inadequate and should stand alone in explaining what’s in the table or figure i.e. Fig 1 Scores of body condition should be something like Fig 1 . average of Condition scores on 50 farms (Number of sheep) measured on ewes and lambs at (when)
- In conclusion mention is made of respiratory diseases in lambs but I could find no data in paper describing that or how it was done in material and methods section
Author Response
the answers are in the attached file

Round 2
Reviewer 1 Report
Thank You for the revised manuscript which has improved from the latter version. The authors have elaborated on some of the comments provided in the first review, while some areas are still unclear.
General comments:
1. The Material & Methods still need additional revision in order to provide the necessary information of the study performed.
-
- Population information as well as the on-farm sample size calculation has been added. However, there is no reference to the included sample of herds being representative. How were herds selected (sample size, sampling strategy/random or convenience)?
- It is still unclear how many observers participated in the data collection
- Section 2.2 Thank You for including justification of the included measures. Regarding the sampling of sheep did farmers select ewes to be scored or were the animals randomly selected?
2. In the Results Section there are no references in the text to tables 2 and 3? Why does Table 1 come in between the figures?
-
- Please add missing references for tables and check caption and legends - Figure 1 is supposed to reflect BCS but the legend says "Cleanliness Score"
- Caption need more information for the tables and figures to stand alone (e.g. "average BCS of Noire de Thibar sheep investigated in 52 Tunisian extensive sheep herds"
- Shifting between past and present tense confuses the reader, please chose one and use it consistently
3. The Discussion of findings is very sparse as it mostly compares findings of the present study with previously reported findings, but no reflections are made.
-
- BCS - what was the variation between herds in your study? Could you see a normal distribution of scores or did some herds have extreme values? Could parasitic infections also be a factor?
- Cleanliness - sheep in the present study are kept on pasture all year yielding a higher cleanliness score - so this is a welfare benefit for sheep, but what do studies with sheep also kept indoors report?
- Health - a discussion of risk factors for sheep lameness under local conditions are missing? What are the challenges - infections (i.e. foot rot) or injuries? The same discussion is missing for respiratory disorders.
- HAR discussion is well written. However, does this relationship differ between extensive and intensive systems?
4. Conclusions - Concluding that HAR is a major determinant of sheep welfare is a bit misplaced here, since you did not test this in the present study. You can, however, conclude whether this HAR was good/better/worse in the studied herds.
Other comments:
- L25 "...scores varied..."
- L28 missing .
- L48-49 The sentence ends with a comma, so there is something missing. Suggest to add: "...sheep using selected aniaml-based welafre measures."
- L67 New paragraph, change verbs to past tense "were"
- L85, L170 past tense
- L123 What does "rear" cover and why is it mentioned here and nowhere else in the text?
- L171 "respiratory rate"
- L172 missing .
- L173 "flocks"
- L177 missing space "In this study the presence..."
- L180 "...a novel person..."
- Table 2 legend: un -> in
- L201 "...on-farm welfare assessment..."
Author Response
Responses to reviewer 1 :
General comments:
- The Material & Methods still need additional revision to provide the necessary information of the study performed.
- Population information as well as the on-farm sample size calculation has been added. However, there is no reference to the included sample of herds being representative. How were herds selected (sample size, sampling strategy/random or convenience)?
- They were selected through random sampling from lists of farmers' data obtained from the Northwest Forestry and Pastoral Development Office of Beja, Tunisia. The farmers were contacted and asked whether they wanted to participate in the study.
- and managed under commercial conditions.
- This number was supported by the AWIN sheep protocol, which recommends a sample of 92 animals when the farm size is ≥ 2000 breeding ewes [10].
- It is still unclear how many observers participated in the data collection
- and examined by a one trained person
- Each farm was visited three times during the period of the study.
- Section 2.2 Thank You for including justification of the included measures. Regarding the sampling of sheep did farmers select ewes to be scored or were the animals randomly selected?
- randomly selected using systematic random sampling and examined by a one trained person at mid-lactation and weaning. Each farm was visited three times during the period of the study.
- In the Results section, there are no references in the text to tables 2 and 3? Why does Table 1 come in between the figures?
- In the text, the references were added
- Please add missing references for tables and check caption and legends - Figure 1 is supposed to reflect BCS but the legend says "Cleanliness Score"
- Modified in the text
- Caption need more information for the tables and figures to stand-alone (e.g. "average BCS of Noire de Thibar sheep investigated in 52 Tunisian extensive sheep herds"
- Changed in the text
- Shifting between past and present tense confuses the reader, please chose one and use it consistently
Changes were made in the text
- The Discussion of findings is very sparse as it mostly compares findings of the present study with previously reported findings, but no reflections are made.
- BCS - what was the variation between herds in your study? Could you see a normal distribution of scores or did some herds have extreme values? Could parasitic infections also be a factor?
- Cleanliness - sheep in the present study are kept on pasture all year yielding a higher cleanliness score - so this is a welfare benefit for sheep, but what do studies with sheep also keep indoors report?
- Health - a discussion of risk factors for sheep lameness under local conditions is missing? What are the challenges - infections (i.e. foot rot) or injuries? The same discussion is missing for respiratory disorders.
- HAR discussion is well written. However, does this relationship differ between extensive and intensive systems?
- The study concerned only an extensive system
- AHR was a good indicator of The stock person's attitudes towards farm animals. Moreover, the low average approach distance our study was explained by the fact that sheep recognize individual humans and are more likely to approach those who treat them well than those who act aggressively. Then, the presence of a familiar human may calm the animals in potentially aversive situations. HAR was a major determinant of sheep welfare and particularly pertinent to extensive systems with limited interactions with people.
- Conclusions - Concluding that HAR is a major determinant of sheep welfare is a bit misplaced here since you did not test this in the present study. You can, however, conclude whether this HAR was good/better/worse in the studied herds.
Other comments:
Added in red color in the text
Reviewer 2 Report
Re paper Assessment of Dairy – Sheep Welfare
Comments – the numbers 1 – 10 refer to my original review
1 Some of my queries have been answered others have not
1.Ok
- as title is dairy sheep there needs to be comment that the Noire de Thibar sheep is a dairy breed and a comment that these farmers were dairy sheep farmers or are they something else?
- It is still not clear how often the farms were visited was it two occasions ie mid lactation and weaning and hence two observations and collection of data but data in graphs indicate only one observation
4 you have indicated measurements were taken by a trained person was it the same person on all 52 farms on all occasions if not how did you standardize between operators – this needs to be made clear in material and methods
- thanks for including standard errors
- still have not made it clear if it was same person on each farm on all occasions
- thanks that is now clear
- so were measurements taken over one hour or was it the mean of a certain time period please be clear what you did here
9 the table and figure headings are still inadequate they need to explain fully what is in the table
Note figure one is incorrectly labelled cleanliness score when it is condition score
And correct title for figure 1 should read
Average condition scores for ewe and lamb taken on 52 farms at mid lactation and at weaning. (although from the graph it looks like only one occasion)
Please include all detail in captions for rest of tables and graphs
- The point about respiratory diseases of lambs was not answered in fact in results only measurements in the results given for lambs was condition score and cleanliness
Further new comments
- Lines 28 -29 “ten farm has fewer than 7% lambs with a low conformation” I could not see any conformation scores just condition scores!!
- Lines 25 -26 the human animal relationship score was not counted on a 1 -3 scale or at least not as was reported in your results and methods, so this statement does not make sense. Is it referring to table 3 please explain what this is ?
- Line 54 mention of calves surely the sheep had lambs !!! how many were ewes and how many lambs were in total
- Lines 57 59 Animals grazed natural rangelands (was it all day and night) and when and how often were supplements fed this needs to be written in paper as it alters sheep behavior and potentially results
- Line 105 SD or was standard error rather than standard deviation please explain what was used
Author Response
Reviewer 2
Comments – the numbers 1 – 10 refer to my original review
1 Some of my queries have been answered others have not
1.Ok
- as the title is dairy sheep there needs to be the comment that the Noire de Thibar sheep is a dairy breed and a comment that these farmers were dairy sheep farmers or are they something else?
I am Sorry, I was confused between the two papers because I have actually another work on dairy sheep. In Tunisia, the Noire de that is for meat production, and Sicilosarde was used for milk production in the same region of our study
Assessment of Meat-type Sheep Welfare Using Animal-Based Measures
- It is still not clear how often the farms were visited was it two occasions ie mid-lactation and weaning and hence two observations and collection of data but data in graphs indicate only one observation
They were selected through random sampling from lists of farmers' data obtained from the Northwest Forestry and Pastoral Development Office of Beja, Tunisia. The farmers were contacted and asked whether they wanted to participate in the study.
Each farm was visited three times during the period of the study.
4 you have indicated measurements were taken by a trained person was it the same person on all 52 farms on all occasions if not how did you standardize between operators – this needs to be made clear in material and methods
and examined by a one trained person
- thanks for including standard errors
- still have not made it clear if it was the same person on each farm on all occasions
- and examined by a one trained person
- thanks that are now clear
- so were measurements taken over one hour or was it the mean of a certain time period please be clear about what you did here
9 the table and figure headings are still inadequate they need to explain fully what is in the table
Note figure one is incorrectly labeled cleanliness score when it is condition score
Replaced in the text
And the correct title for figure 1 should read
Average condition scores for ewe and lamb taken on 52 farms at mid-lactation and weaning. (although from the graph it looks like only one occasion)
Replaced in the text
Please include all detail in captions for the rest of the tables and graphs
- The point about respiratory diseases of lambs was not answered in fact results only measurements in the results given for lambs was condition score and cleanliness
Further new comments
- Lines 28 -29 “ten farms have fewer than 7% lambs with a low conformation” I could not see any conformation scores just condition scores!!
body condition scores
- Lines 25 -26 the human-animal relationship score was not counted on a 1 -3 scale or at least not as was reported in your results and methods, so this statement does not make sense. Is it referring to table 3 please explain what this is?
Replaced in the text
- Line 54 mentions calves surely the sheep had lambs!!! how many were ewes and how many lambs were in total
- A total of 1040 Noire de Thibar ewes (840) and calves (200),
- Lines 57 59 Animals grazed natural rangelands (was it all day and night) and when and how often were supplements fed this needs to be written in the paper as it alters sheep behavior and potentially results
The ewes were managed under extensive conditions, in a year-round outdoor system, grazing pastures, and managed under commercial conditions. Natural rangelands forage (hay, barley in green, etc.) and concentrated feed were the main food resources for sheep
- Line 105 SD or was standard error rather than standard deviation please explain what was used
The standard deviation (SD) measures the amount of variability, or dispersion, from the individual data values to the mean, while the standard error of the mean (SEM) measures how far the sample mean (average) of the data is likely to be from the true population mean.
Round 3
Reviewer 1 Report
Most of the comments have been addressed, however, there are still some questions left to be addressed and a thorough proof reading is necessary to correct typo´s, missing spaces between words etc..
Detailed comments:
SAMPLING: Selection of herds and animals - how did you arrive at the 52 herds? Please provide information on this sample being representative of the target population.
In lines 75-77 it still sounds like the farmers are choosing the animals for assessment. If this was the case, you should address this potential selection bias in the discussion.
HAR or AHR? The abbreviations are used interchangeably in the manuscript. In order to improve readability only one term should be used consistently.
Figure and Tables: Captions are still to sparse to explain the given data, Legends and captions should be proof-read (i.e. use of capital letters)
Discussion: Since ewes were assessed at three occasions, when did you observe most thin ewes? Can these findings be related to the production cycle? How do helminths affect sheep in extensive systems?
Cleanliness - since cleanliness scores found in the present study were higher than previously reported, it would be good to have an idea of how the present animals were housed? There is no real discussion, just a comparison to other´s findings. This is also the case for the health parameters.
Author Response
Dear Editor
Please in the attached file the response to reviwer1

Reviewer 2 Report
Re paper Assessment of Dairy – Sheep Welfare Review 3 !!!
Comments – some queries in previous review number 2 were still not answered adequately
- Lines 60 -62 It is still not clear how often the farms were visited was it two occasions ie mid lactation and weaning and hence two observations and collection of data but data in graphs indicate only one observation and in text ITS says body condition scores taken at lambing so was that the third time whereas in text just says mid lactation ands weaning
- Average condition scores for ewe and lamb taken on 52 farms at (please record when it the measurements were taken ie lambing mid lactation or weaning which one was it) from the graph it looks like only one occasion and line 119 IT says taken at lambing
- Please include all detail in captions for rest of tables and graphs ie when were readings recorded
- The point about respiratory rate of lambs was not answered in fact in results only measurements in the results given for lambs was condition score and cleanliness and respiratory rates in table 2 are stated for ewes only
- Line 58 mention of calves surely the sheep had lambs!!!

Author Response
Dear Editor
In attached file the responses required
Cordially
